# Regulated Ecosystem Services Trade-Offs: Synergy Research and Driver Identification in the Vegetation Restoration Area of the Middle Stream of the Yellow River

Ge Wang, Depeng Yue *, Teng Niu and Qiang Yu 

Beijing Key Laboratory for Precision Forestry, Beijing Forestry University, Beijing 100107, China; GlennW0912@bjfu.edu.cn (G.W.); niuteng21@bjfu.edu.cn (T.N.); yuqiang@bjfu.edu.cn (Q.Y.)
\* Correspondence: yuedepeng@126.com or yuedepeng@bjfu.edu.cn; Tel.: +86-010-6233-7585

**Abstract:** In arid and semi-arid regions, vegetation restoration will have a significant impact on ecosystem services (ESs). Accurate assessment of the relationship and driving mechanism between ESs will play an important role in the implementation of subsequent vegetation restoration projects and ecosystem management. The purpose of this study is to evaluate and identify the relationship between ESs, and explore the impact and driving mechanism of ecological restoration on the relationship between ESs. Taking the middle reaches of the Yellow River as the study area, this study analyzed three ESs including: Net Primary Productivity (NPP), soil conservation (SC), and water yield (WY), in the vegetation restoration area (VRA) in the middle reaches of the Yellow River for 20 years (2000–2010 and 2010–2020 years(a)). Taking the ecological restoration unit (ERU) as the evaluation unit, we evaluated the impact of three vegetation restoration models on ESs. Using geo-detectors to identify the economic, social and natural drivers that affect the relationship between ESS. The results showed the following: (1) Different vegetation restoration models will lead to significant differences in the restoration rate for ESs. They will lead to an increase in the recovery rate of NPP and SC. The first vegetation restoration mode and the third planting restoration mode will reduce the restoration rate for the WY. (2) The three vegetation restoration models will enhance the synergistic relationship between NPP and SC. They will weaken the trade-off relationship between NPP and WY, SC and WY. (3) Temperature, precipitation, and NDVI will affect the changes in ESs in VRAs. The ESs trade-off-synergy relationship will be affected by precipitation, NDVI, and GDP. This study showed that, with the implementation of vegetation restoration projects, ESs in time and space and the complex heterogeneity of ESs will affect ecosystem management. The results of this study will be helpful for the implementation of subsequent vegetation restoration projects and provide scientific advice for ecosystem management.

**Keywords:** vegetation restoration; ecosystem service; ecological restoration unit; geo-detector; the middle reaches of the Yellow River

## 1. Introduction

Ecosystem services (ESs) are many benefits that healthy ecosystems bring to humans, including supply services, regulatory services and cultural services [1]. ESs are an important link between ecosystems and human well-being. There are many types of ESs, and their spatial distribution varies greatly [2]. At the same time, human beings selectively use ESs, which leads to the dynamic changes of the relationship between ESs, which is manifested in the trade-offs of mutual constraints or mutually beneficial synergy [3]. ESs are intertwined, and the interaction relationship is highly nonlinear. Strengthening the understanding of the highly nonlinear relationship between ecosystem and quantitative expression is a prerequisite for a more scientific evaluation of ecosystem value [4]. Especially in arid and semi-arid areas, vegetation restoration will have a significant impact on the service

function of regulated ecosystem. It can improve the effect of ecosystem management based on regulated ecosystem services (RESs) and realize the sustainable supply of RESs [5].

In order to study the temporal and spatial changes of ESs and the relationship between ESs, scholars at home and abroad have adopted a large number of methods [6]. At present, commonly used research methods include graphic comparison, scenario analysis and model simulation [7]. Spatially, the superposition analysis of ESs can determine the type and scope of trade-off coordination among ESs [8]. Egoh et al. spatially superimposed five ESs (surface water supply, water flow regulation, carbon storage, soil accumulation and soil conservation) and biodiversity hotspots, concluded that the correlation between the five ES functions and biodiversity was weak. In scenario analysis, if many scenarios are formulated, such as ecological protection priority, socioeconomic development priority or both, the dynamic changes between various ESs were analyzed [9]. Shui et al. simulated the spatial distribution and trade-off synergy of four kinds of ESs in Fujian Delta Urban Agglomeration in 2030 under three scenarios (NATURAL scenario, planning scenario and protection scenario). The results showed that the synergy of protection scenarios was higher and the trade-off relationship was lower. However, in the natural scenario, the number of ESs decreases significantly, the synergy relationship shows a downward trend, and the trade-off relationship intensifies [10]. Model simulation used ES mechanism or statistical model to simulate the service volume of different ecosystems, and conduct trade-offs and collaborative analysis. Since the implementation of the Yellow River Basin reconstruction project in 1999, Bai et al. used SWAT software to analyze the relationship between drought, soil and water conservation services, meteorology, vegetation and other factors. Since ecological restoration, the relationship between soil conservation and water production services had been a trade-off. The function of SC had been improved, but the drought had intensified [11].

Ecological restoration is a process that helps restore degraded, damaged or damaged ecosystems [12]. Ecological restoration can change the patterns and processes of ecosystems and promote the restoration of RESs [13]. It can affect the evolution of regulatory ESs and the synergy and trade-off between different RESs [14]. Vegetation restoration is an important way of terrestrial ecosystem restoration. It is the process of planting and configuration of plants, plant community restoration or reconstruction, or natural renewal and restoration of a plant community [15]. As one of the most widely implemented and largest vegetation restoration projects in the world, China's farmland to forest project (GFGP) has attracted extensive attention all over the world [16]. GFGP aims to restore cultivated land on steep hillsides to forest land or grassland, and restore barren mountains and wasteland suitable for tree growth so as to alleviate and prevent flood and soil erosion [17]. The middle stream of the Yellow River is a pilot and demonstration area for the project of returning farmland to Forests [18]. The conversion of farmland to forest project has changed the original land use type of vegetation restoration area (VRA) and had a significant impact on its ESs [19]. A large number of previous studies have shown that the project of returning farmland to forests reduces soil erosion and increases carbon sequestration on the middle stream of the Yellow River [20]. However, newly added vegetation may lead to reduced water production services and exacerbate drought in some areas [21]. On different scales, the implementation of ecological restoration may lead to opposite changes in ES functions, while the enhancement of one or more ES functions may lead to the reduction of other ES functions [22]. In order to realize the sustainable supply of multiple ESs, it is necessary to accurately analyze the relationship between multiple RESs and formulate more scientific management policies [23].

The management of multiple RESs, especially the relationship between balanced RESs, is a hotspot in the field of ecological restoration and ESs [24]. Chen et al. concluded that there are trade-offs and synergies between supply, regulation and cultural services. Large-scale ecological restoration projects are one of the key drivers of changes in the relationship between ESs [25]. Feng et al. used redundancy analysis to explore the impact of environmental factors on the trade-off synergy relationship between ESAs in Ansai

basin [26]. In terms of time and space, Sun et al. studied the change characteristics of the trade-off and coordination relationship between the four ES values (NPP, soil and water conservation, water conservancy and food supply) in Yan'an City. They analyzed the impact of the policy of returning farmland to forest and grassland on the dynamic changes of the trade-offs and synergies of ES [27]. By simulating the scenario, Yang et al. concluded that the trade-off effect is the best when returning farmland to forest and grassland. Vegetation restoration has both positive and negative effects on the ecosystem. At present, under the background of the policy of "ecological protection and restoration of mountains, rivers, forests, farmland, lakes and grasslands", many areas are still returning farmland to forest and grassland on a large scale. It is particularly important to pay attention to the dynamic changes of RESs and the relationship between regulatory ESs in the Loess Plateau, and accurately evaluate the relationship between RESs. This can provide scientific basis for regional ecological restoration and ecosystem management [28].

The study of the impact of vegetation restoration on the relationship between RESs is of great significance for the implementation of subsequent ecological restoration projects and ecosystem management [29]. At present, the research on the middle stream of the Yellow River ESs mainly focuses on a single time node or a single ESs [30]. With regard to the impact of vegetation restoration on the function of RESs, this study focuses on evaluating the trade-off synergy of regional ESs after restoration [31]. Most of the previous studies lack the comparative study of different vegetation restoration methods before and after restoration. In addition, it is easy to ignore the scale effect in the research. The evaluation based on the trade-off and coordination relationship of pixel or regional ESs, taking the administrative region as the unit, has no direct guiding significance for the follow-up ecological restoration and ecosystem management. Based on the county scale, this study puts forward the concept of ecological restoration unit (ERU), which connects the evaluation of ESs with county-level administrative regions, and fills this research gap.

At the same time, on the ERU scale, the changes of ES before and after vegetation restoration and the changes of balance and coordination relationship of ES were compared. Due to the spatiotemporal heterogeneity of ES, its dynamic changes under the influence of vegetation restoration were more complex. We used geographic detectors to identify the economic, social and natural driving factors affecting vegetation restoration, so as to provide support for returning farmland to forest and grassland in the future. The purpose of this study is: (a) to establish an ES evaluation framework suitable for county scale; (b) study the influence of different vegetation restoration methods on the change of ES and the relationship between ESs; (c) under the leadership of different vegetation restoration methods, quantify the impact of economic, social and environmental factors on vegetation restoration.

## 2. Materials and Methods

### 2.1. Study Area

The study area is located in the middle reaches of the Yellow River in China (Figure 1). The geographical position is between 32° and 42° north latitude and between 104° and 113° east longitude. The total area is about 343,845 km$^2$, and the area is about 48.1% of the area of the Yellow River Basin. It is also an important part of the Loess Plateau, covering an area of about 53% of the area of the Loess Plateau, with an elevation ranging from 12 to 3936 m, and the terrain decreases from northwest to southeast. The length of the main stream of the Yellow River in the study area is about 1235 km [32]. The study area has a temperate arid and semi-arid climate, with annual precipitation varying from 300 to 800 mm [33]. The landform is dominated by loess hilly landforms. The soil is loose and easily eroded. It is an area with more serious soil erosion in the Yellow River Basin [34]. Its erosion and sediment yield are the main sources of siltation and sedimentation in the lower reaches of the Yellow River [35]. Since 1999, a large-scale ecological restoration project of returning farmland to forest and grassland has been fully implemented in the study area. The ecological environment has gradually recovered. At the same time, there are new problems, such as the continuous

decrease in the runoff of the mainstream of the Yellow River [36]. Therefore, since 2016, in order to ensure the ecological environment and that the quality of safety barriers promotes the virtuous cycle and sustainable use of the ecosystem, the pilot project of "the Ecological Protection and Restoration of Mountains–Rivers–Forests–Farmlands–Lakes–Grasslands" has been gradually implemented in the study area [37].

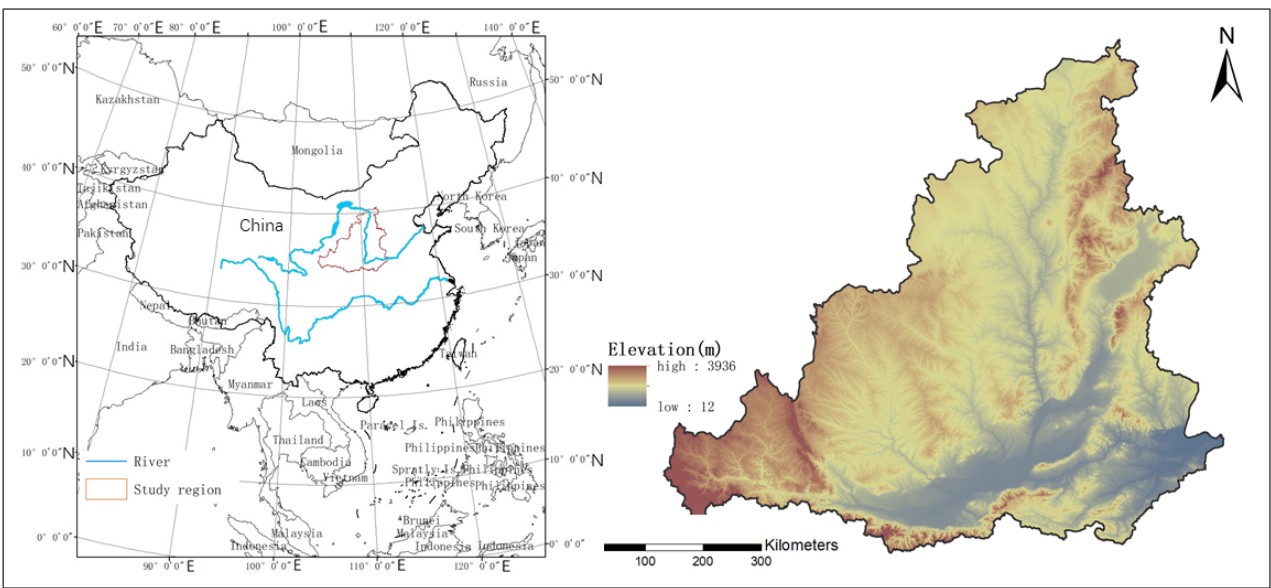

**Figure 1.** Study area.

### 2.2. Data Source

The following data sets were used in this study:

1. Thirty-meter global land cover data from GlobeLand30: Global Geo-information Public Product land use data of the study area in 2000, 2010, and 2020, including farmland, woodland, grassland, shrubland, wetland, water, artificial surface, and bare land.
2. China Meteorological Data Network provides the daily sunshine duration (h), temperature (°C), and precipitation (mm) data.
3. The official website of the United States Geological Survey (USGS) provides MODIS data, including MOD13Q1 (Normalized Vegetation Index NDVI), MOD16A2 (Evapotranspiration and Potential Evapotranspiration data), and MOD44 (tree coverage rate and non-tree vegetation coverage rate). Geo-spatial Data Cloud provides the Digital Elevation Model (DEM) data of the study area.
4. The official website of the National Qinghai-Tibet Plateau Science Data Center provides the soil data of the study area, including the sand content, silt content, clay content, and organic carbon content of the soil.
5. Statistical Yearbooks issued by Shanxi Provincial Bureau of Statistics, the Shaanxi Provincial Bureau of Statistics, the Henan Provincial Bureau of Statistics, the Gansu Provincial Statistics Bureau, the Inner Mongolia Autonomous Region Statistics Bureau, and the Statistics Bureau of Ningxia Hui Autonomous Region provided GDP and population data of the counties and cities where the study area is located.

The important data are summarized in Table 1.

**Table 1.** Important data.

| Data | Time Resolution | Spatial Resolution | Unit |
|---|---|---|---|
| Land-Use and Land-Cover Change (LUCC) | 1 year | 30 m × 30 m | |
| Sunshine duration | 1 day | | H |
| Air temperature | 1 day | | °C |
| precipitation | 1 day | | mm |
| Normalized vegetation index NDVI | 8 day | 250 m × 250 m | |
| Evapotranspiration and potential evapotranspiration | 16 day | 500 m × 500 m | |
| Tree coverage and non-tree Vegetation coverage | 1 year | 1000 m × 1000 m | |
| DEM | | 30 m × 30 m | m |
| Sand content, silt content and clay content | | 1000 m × 1000 m | % |
| GDP | 1 year | | 10,000 yuan |
| Population | 1 year | | One people |

*2.3. Land Use Change*

The GlobeLand30 data in 2000, 2010, and 2020 were used to analyze the changes in land-use before and after the implementation of ecological restoration (the land-use types in the study area are divided into cultivated land, woodland, grassland, shrubland, wetland, water body, human-made surface, and bare land). The overall accuracy of GlobeLand30 data in 2000 and 2010 is 83.50%, and the Kappa coefficient is 0.78; the overall accuracy in 2020 is 85.72%, and the Kappa coefficient is 0.82. Therefore, these data meet the needs of land use change analysis in the study area. The land-use transfer matrix was used to analyze the land-use conversion relationship between 2000–2010 and 2010–2020, and the results of land-use change were listed in matrix form [38]. Based on the results of land use change analysis, three parts of new vegetation were extracted (including forest land or grassland converted from cultivated land and woodland converted from grassland) [39].

*2.4. Estimation of Ecosystem Services*

Vegetation restoration can have a significant impact on RESs. Based on the ArcGIS platform, the net primary productivity (NPP), soil conservation (SC) and water yield (WY) of VRAS in 2000, 2010 and 2020 were evaluated using the investment model. Invest model solves the problems of unclear formation mechanism of ES function and poor evaluation results. It has gradually become the most widely used ecosystem service evaluation model [40]. By quantifying the ecological service function under ecological restoration, we can better clarify the interaction mechanism between ESs. At the same time, the impact of vegetation restoration on ESs can be clarified [41].

2.4.1. Plant Net Primary Productivity

The NPP refers to the total amount of dry matter accumulated by green plants per unit time and area [42]. It can characterize the activities of plants themselves. It plays an important role in the terrestrial carbon cycle, the balance of carbon dioxide and oxygen, and the regulation of global temperature [43]. In the study area, especially in areas where vegetation is restored, an increase in NPP can improve regional gas regulation and climate regulation services. The Carnegie–Ames–Stanford Approach (CASA) method is used to estimate NPP. NPP $(C/MJ)$ is mainly determined by the solar radiation (APAR: $\frac{MJ}{m^2}.a$) absorbed by plants and the light energy conversion rate ($\varepsilon$: $gC/MJ$):

$$NPP = APAR * \varepsilon \tag{1}$$

The solar radiation absorbed by plants is determined by the total solar radiation (SOL: $MJ/m^2$) and the effective light absorption ratio (FPAR) of vegetation:

$$APAR = SOL * FPAR * 0.5 \tag{2}$$

The light energy conversion rate is the ratio of the solar radiation absorbed by the vegetation into organic carbon. In Formula (3), $T_{\varepsilon 1}$ represents the limiting factor of plant photosynthesis at high and low temperatures, and $T_{\varepsilon 2}$ represents the time cooperation of plants changing from suitable temperature to high or low temperature. $\varepsilon^*$ is the maximum light energy conversion rate in an ideal state. According to Potter's research results, the value of $\varepsilon^*$ is 0.389 $gC/MJ$. The formula for the light energy conversion rate is

$$\varepsilon = T_{\varepsilon 1} * T_{\varepsilon 2} * W_\varepsilon * \varepsilon^* \tag{3}$$

2.4.2. Soil Conservation

The study area is located in the soil erosion intensive area of the Yellow River Basin [44]. Estimating the amount of soil conservation is a prerequisite for improving the soil conservation service function in this area. Based on the Surface Cover Replacement Hypothesis, the modified soil erosion equation (ULSE) is used to estimate the amount of potential soil erosion and the actual amount of soil erosion, and the difference between the two is the amount of soil conservation in the ecosystem [45]. Potential soil conservation refers to the amount of soil erosion that the ecosystem may produce without vegetation cover and soil conservation measures, that is, $C = 1$, $P = 1$; the actual soil conservation refers to the existing vegetation coverage and soil conservation measures below. The amount of soil erosion produced by the ecosystem and the amount of soil conservation is

$$SC = R \times K \times LS \times (1 - C \times P) \tag{4}$$

$SC$ is the amount of soil conservation in the ecosystem($t/hm^2/a$); $R$ is the rainfall erosivity factor; $K$ is the soil erodibility factor; $LS$ is the slope length and slope factor; $C$ is the vegetation coverage factor; $P$ is the soil conservation measure factor.

2.4.3. Water Yield (WY)

The study area is located in the Loess Plateau, in an arid and semi-arid area, with scarce water resources and less surface runoff [46]. Based on the theory of water balance, water production in arid and semi-arid areas is mainly determined by annual rainfall (*AR*: mm) and evapotranspiration (*ET*: mm) [47]. The calculation formula for water production is

$$WY(mm) = AR - ET \tag{5}$$

The precipitation data are provided by 61 meteorological stations in and nearby the study area. The precipitation of each day of each weather station is added to obtain the annual precipitation, and the inverse distance weighting method is used for interpolation. The evapotranspiration data comes from MOD16A2 (ET), which has a high consistency with the measured value of ET in forest areas. The accuracy is about 0.76 in several major river basins in China, which meet the accuracy requirements of this study.

*2.5. Evaluation Framework of Ecosystem Services in Vegetation Restoration Areas*

In this study, the county-level administrative unit vector boundary was used to divide the study area into 226 sub-areas. Each sub-region was called an ecological restoration unit (ERU) (Figure 2). At the scale of the ERU, we studied the impact of vegetation restoration on ESs. The vegetation restoration mode (VRM) referred to the integration of technologies that use one or more plant species to promote vegetation restoration in areas with severe soil erosion for the purpose of ecological restoration, taking into account economic and social benefits, and using one or more plant species. Under the background

of the Grain for Green Project, the study area has experienced vegetation restoration for nearly 20 years, and there are three types of VRMs (Figure 2): the first and second (first VRA and second VRA) were aimed at sloping lands that cause soil erosion and arable land that was prone to land desertification, and the third (third VRA) was for barren hills and wastelands suitable for tree growth. The three types of VRMs correspond to the three types of vegetation restoration areas (VRAs), which were divided into ERUs in ArcGIS, and the area of vegetation restoration and ecosystem services in each ERU was counted.

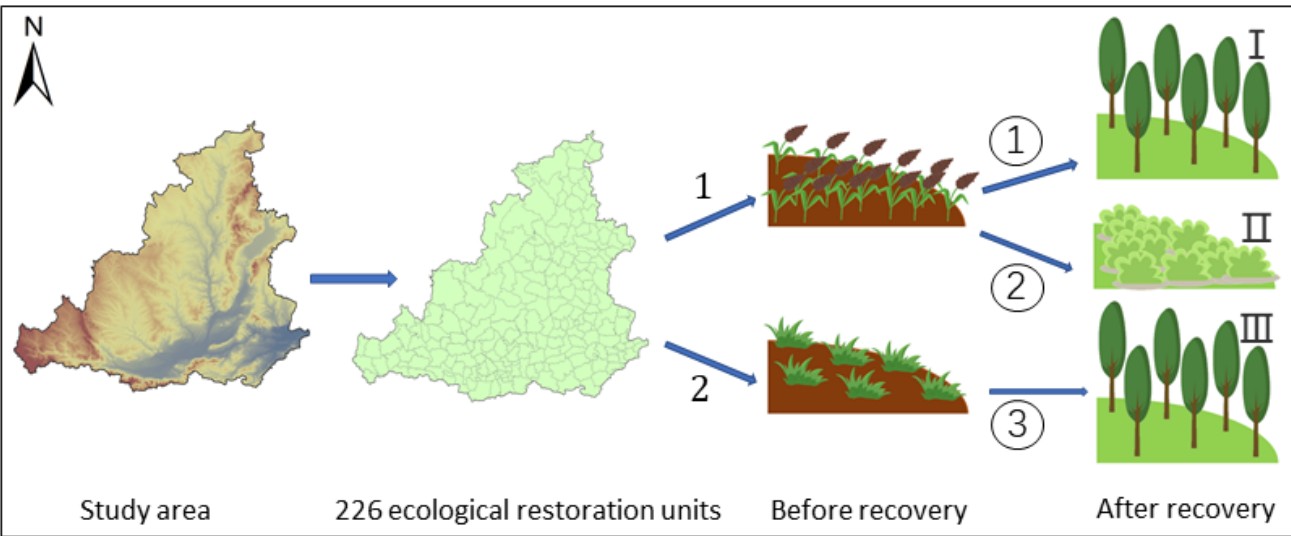

**Figure 2.** The framework of vegetation restoration in the middle reaches of the Yellow River. Before restoration, 1 was the sloping farmland with serious soil erosion, 2 was the barren hills and wasteland suitable for planting trees, and the land use type was mostly grassland. 1 refers to the first VRM, where the cultivated land was restored to forest land, and I was the first VRA; 2 refers to the second VRM, where the cultivated land was restored to grassland, and II was the second VRA; 3 refers to the third VRM, where the grassland was restored to woodland, and III was the third VRA.

Based on the ERU, the ecosystem service restoration rate was used to evaluate the restoration effect of ecosystem services in the study area [48]. The ecosystem service restoration rate was the ratio of the amount of ecosystem service change to the amount of ecosystem service after restoration. The formula is

$$ES_{\text{recove}} = \frac{ES_{\text{After}} - ES_{\text{Before}}}{ES_{\text{Before}}} \times 100\% \tag{6}$$

$ES_{\text{recove}}$ is ecosystem service restoration rate; $ES_{\text{After}}$ is ecosystem service value before ecological restoration; $ES_{\text{Before}}$ is ecosystem service value after ecological restoration.

Due to the large variety of ESs, their differences in spatial distribution, and the influence of human factors, the relationship between ESs has undergone a dynamic change, which is manifested in the trade-offs and synergy of mutual gains. The trade-off is that one ES decreases due to the increase of other ESs. Synergy is manifested in the simultaneous increase of two or more ecosystem services. The relationship between ESs can be determined through trade-offs and collaborative analysis. The result can be used as a guide for ecosystem protection. It can realize the continuous output of ES products and provide well-being for human society. Pearson correlation was used to analyze the correlation between the two ecosystem service functions, and the correlation test was carried out ($p < 0.05$). A positive correlation coefficient indicates that there is a synergistic relationship between ESs, and a negative correlation coefficient indicates that there is a trade-off relationship.

*2.6. Geo-Detectors*

Changes in ecological service functions of VRAs may be closely related to certain environmental or economic and social factors [49]. Changes in ecological service functions may be the result of a combination of multiple factors. Many factors interact regarding ecosystem services, and it is difficult to quantify the degree of influence of a certain factor on ecosystem services. In the research of spatial hierarchical heterogeneity and variable classification, geographic detectors can provide technical support for the identification of impact factors. Geographic detectors have unique advantages in multi-factor interactive recognition. In recent years, it has been gradually applied in the fields of land management, landscape ecology, urban and rural planning, etc. The core idea is based on the assumption that, if an independent variable has an important influence on a dependent variable, the spatial distribution of the independent variable X and the dependent variable Y should be similar [50]. The correlation between X and Y can be explained by the q-value. The formula is

$$q = 1 - \frac{\sum_{h=1}^{L} N_h \sigma_h^2}{N \sigma^2} \tag{7}$$

where $h = 1, \ldots, L$ is the stratification of variable Y or factor X (Figure 3); $N_h$ and N are the number of units in layer $h$ and the whole area, respectively; $\sigma_h^2$ and $\sigma^2$ are the value of layer $h$ and Y-value in the whole area, respectively.

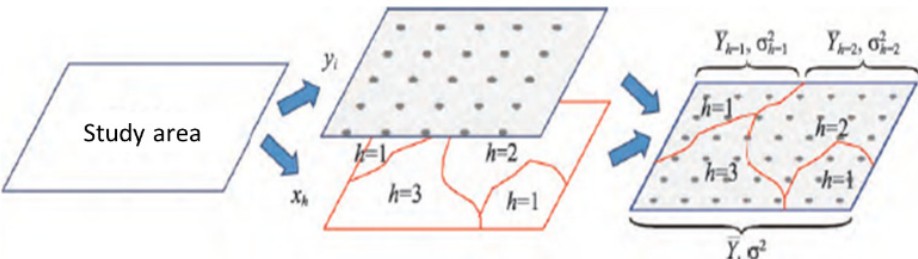

**Figure 3.** Schematic diagram of the principle of geographic detectors.

## 3. Results

### 3.1. The Impact of Vegetation Restoration on Land Use Types

Vegetation restoration has had a significant impact on land use in the study area. During 2000–2020*a*, the area of arable land showed a decreasing trend (Table 2), with a decrease of 6124.79 km$^2$. The area of woodland continued to increase, increasing by 2829.94 km$^2$, and the area of grassland was decreasing, which reduced by 4782.33 km$^2$. From 2000 to 2010*a*, 1356.81 km$^2$ and 2744.91 km$^2$ of arable land were converted into woodland and grassland. These were, respectively, 0.92% and 1.87% of the total farmland area. The area converted from grassland to woodland was 3446.87 km$^2$, accounting for 3.48% of the total grassland area. The newly increased vegetation area caused by vegetation restoration accounted for 2.18% of the total area of the study area. During 2010–2020*a*, the area of farmland converted to woodland and grassland was 1887.63 km$^2$ and 5085.63 km$^2$ (Table 3), which are 1.25% and 3.39% of the total farmland area. The area of grassland converted to woodland was 4958.33 km$^2$, accounting for 5% of the total grassland area. The area of land-use change caused by vegetation restoration accounted for 3.46% of the total area of the study area.

**Table 2.** 2000–2010 land use transfer matrix (km$^2$).

| 2010 | 2000 | | | | | | | |
|---|---|---|---|---|---|---|---|---|
| | **Farmland** | **Woodland** | **Grassland** | **Shrubland** | **Wetland** | **Water** | **Artificial Surface** | **Bare Land** |
| farmland | 141,766.75 | 839.02 | 2352.48 | 26.13 | 199.08 | 292.42 | 1299.86 | 10.02 |
| woodland | 1356.81 | 80,750.66 | 3446.87 | 10.03 | 17.85 | 41.07 | 30.32 | 6.43 |
| grassland | 2744.91 | 1004.63 | 92,568.19 | 70.06 | 40.32 | 92.14 | 73.91 | 82.54 |
| shrubland | 30.22 | 6.72 | 263.44 | 999.33 | 1.11 | 4.54 | 0.66 | 0.36 |
| wetland | 145.40 | 4.40 | 34.53 | 0.12 | 157.91 | 92.37 | 0.91 | 0.07 |
| water | 390.21 | 28.41 | 86.17 | 2.11 | 119.93 | 735.19 | 5.72 | 0.22 |
| artificial surface | 3500.53 | 31.53 | 273.09 | 1.18 | 3.54 | 11.00 | 6375.82 | 7.61 |
| bare land | 31.05 | 5.95 | 81.49 | 1.73 | 0.20 | 2.29 | 0.57 | 2783.74 |

**Table 3.** 2010–2020 land use transfer matrix (km$^2$).

| 2020 | 2010 | | | | | | | |
|---|---|---|---|---|---|---|---|---|
| | **Farmland** | **Woodland** | **Grassland** | **Shrubland** | **Wetland** | **Water** | **Artificial Surface** | **Bare Land** |
| farmland | 134,152.10 | 1873.18 | 6047.11 | 88.15 | 85.12 | 250.38 | 1250.41 | 94.64 |
| woodland | 1887.63 | 78,446.67 | 4958.33 | 122.94 | 2.16 | 12.68 | 21.90 | 48.95 |
| grassland | 5085.63 | 4955.56 | 82,945.44 | 534.21 | 11.35 | 48.11 | 72.13 | 671.50 |
| shrubland | 76.16 | 111.83 | 524.70 | 524.24 | 0.08 | 0.94 | 0.28 | 6.02 |
| wetland | 72.07 | 16.76 | 22.26 | 0.05 | 264.31 | 151.54 | 0.52 | 0.33 |
| water | 298.95 | 55.01 | 116.72 | 5.69 | 66.47 | 887.38 | 10.69 | 2.48 |
| artificial surface | 5185.97 | 146.99 | 1371.84 | 13.09 | 2.92 | 11.57 | 8847.94 | 60.38 |
| bare land | 27.23 | 54.04 | 690.30 | 17.99 | 3.30 | 5.37 | 0.42 | 2022.72 |

The study area was divided into 226 ERUs using county boundaries, and three VRAs in each ERU were counted in 2000–2010a and 2010–2020a (Figure 4). During 2000–2010a, due to the different times and effects of the project of returning farmland to forest and grassland, the newly added forest land (the forest land converted from cultivated land and the forest land converted from grassland) was significantly different in space. In the southern part of the study area, Luanchuan, the ecological restoration area in Qingshui and Wushan was relatively large. Due to the implementation of the project of returning farmland to forest, the area of newly increased forest land converted from cultivated land was more than 50 km$^2$. In the ERU in Zhang County, the area of newly increased forest land converted from grassland reached more than 1000 km$^2$. The newly added grassland was mainly concentrated in the central and northern parts of the study area, and the newly added grassland areas in Hengshan District, Shenmu, and Gujiao were more than 80 km$^2$. During 2010–2020a, the spatial distribution of newly added forest land and grassland in the study area underwent major changes compared with 2000~2010a and was mainly concentrated in the west and north of the study area.

### 3.2. Changes in Ecosystem Services

Land use and climate change will promote changes in RESs. In order to analyze the impact of vegetation restoration on RESs, we must first exclude the impact of climate change on RESs. 2000~2010a and 2010~2020a, all ecosystems in the study area experienced the effects of precipitation and temperature changes. However, the project of returning farmland to forest and grassland only affected the VRA. In VRAs, comparing RESs before and after restoration can effectively explain the impact of vegetation restoration on ecosystem services.

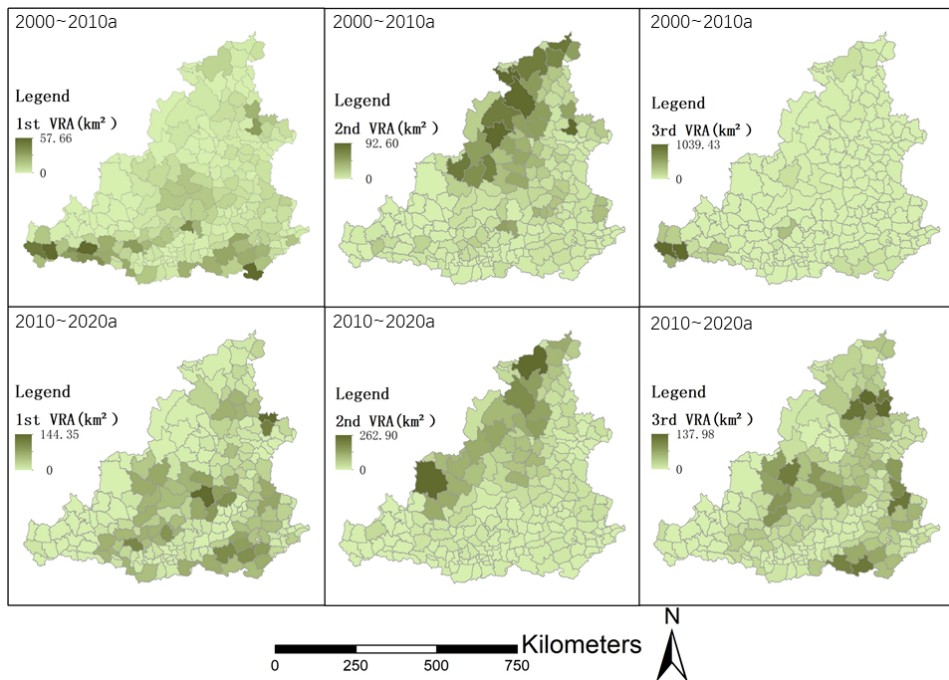

**Figure 4.** Area statistics of the three VRAs in an ecological restoration unit (ERU).

Based on the ERU, the amount of RESs in the three VRAs from 2000–2010a and 2010–2020a was calculated, and the ES restoration rate was calculated (Figure 5). The ratio of the amount of RES change in the area of vegetation restoration to the amount of RES after restoration is defined as the RES restoration rate, which is used to intuitively reflect the impact of vegetation restoration on RESs.

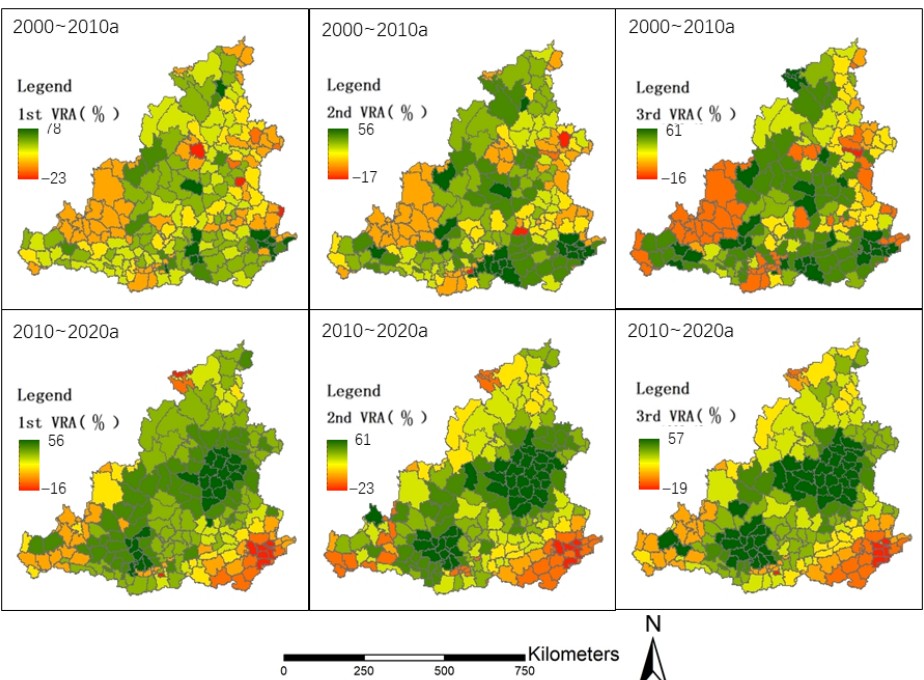

**Figure 5.** NPP restoration rate distribution of 226 ERUs.

During the period from 2000 to 2020a, the total NPP recovery rate of VRA in the study area was greater than zero. The total amount of NPP is on the rise. Vegetation restoration led to an increase in the total NPP in the study area. From 2000 to 2010a, the NPP recovery rate of the first VRA in the study area was the highest, with a recovery rate of 43%; the NPP recovery rate of the third VRA was the lowest, with a recovery rate of 29%. Compared with 2000–2020a in 2010–2020a, the NPP recovery rates of the two VRAs in the study area increased. The third VRA had the highest recovery rate at 51% and the first VRA had the lowest recovery rate at 40%. Unlike the study area scale, at the ERU scale, the NPP recovery rate (Figure 5) has a significant spatial differentiation. In the first phase of 2000–2010a, the high-value areas of VRA NPP recovery were concentrated in the central and southern regions, and the low-value areas were concentrated in the west. The high-value areas of the second-stage VRA NPP recovery are concentrated in the central and southern parts, and the low-value areas are scattered in the eastern and western parts. The high-value areas of the third-phase VRA NPP recovery factor are scattered in the middle, and the low-value areas are in the east and west. During the period from 2010 to 2020a, the spatial distribution characteristics of NPP recovery rates in the three recovery areas were relatively consistent: the high-value areas of NPP recovery rate were concentrated in the central and southern parts, and the low-value areas were concentrated in the south and west.

From 2000 to 2020a, the total SC service recovery rate of the three VRAs in the study area was all greater than 0, and the vegetation restoration led to an increase in the total soil conservation in the study area. During the period from 2000 to 2010a, the recovery rate of total SC in the third VRA was the highest, with a recovery rate of 33%; the total recovery rate of SC in the second VRA was the lowest, with a recovery rate of 23%. From 2010 to 2020a, the recovery rate of SC in the second VRA was the highest, with a recovery rate of 34%; the recovery rate of SC in the third VRA was the lowest, with a recovery rate of 15%. At the ERU scale, the spatial analysis of SC recovery rates was significant (Figure 6). From 2000 to 2010a, in the first VRA, high-value units were concentrated in the north and south, and low-value units were concentrated in the north and east. The high-value cells of the secondary VRA recovery rate were concentrated in the north and middle, and the low-value cells were scattered in the middle and south. The high-value units of the third VRA are scattered across the study area, and the low-value units are concentrated in the north. During 2010–2020a, the spatial distribution of recovery rates of the three VRAs was relatively consistent: high-value cells were distributed in the north and middle, and low-value cells were distributed in the north and southwest. The proportion of high-value units in Phase I VRA is higher than that in other types of recovery areas, and the proportion of low-value units in Phase III VRA is higher than that in other types of VRA.

From 2000 to 2010a, the WY recovery rates of the three VRAs were all negative, and the third VRA had the lowest total WY recovery rate, with a recovery rate of −57%. Vegetation restoration resulted in increased water use in the restoration area. WY is trending down and has not recovered. From 2010 to 2020a, the second VRA had the highest WY with a total WY recovery rate of 38%. Unlike at the regional scale, WY restoration rates showed clear spatial differentiation at the ecological unit scale. During the period from 2000 to 2010a, the low-value units of the first VRA WY recovery factor were widely distributed (Figure 7), and the high-value units were distributed in the south and north. The high-value units of the second VRA are concentrated in the north, and the low-value units are distributed in the central and southeastern regions. The high-value units of the third VRA are distributed from north to south. From 2010 to 2020a, the spatial distribution of the WY recovery rates of the three VRAs was relatively consistent: the units with positive WY recovery rates were concentrated in the middle and southwest, and the units with negative WY recovery rates were concentrated in the north and south.

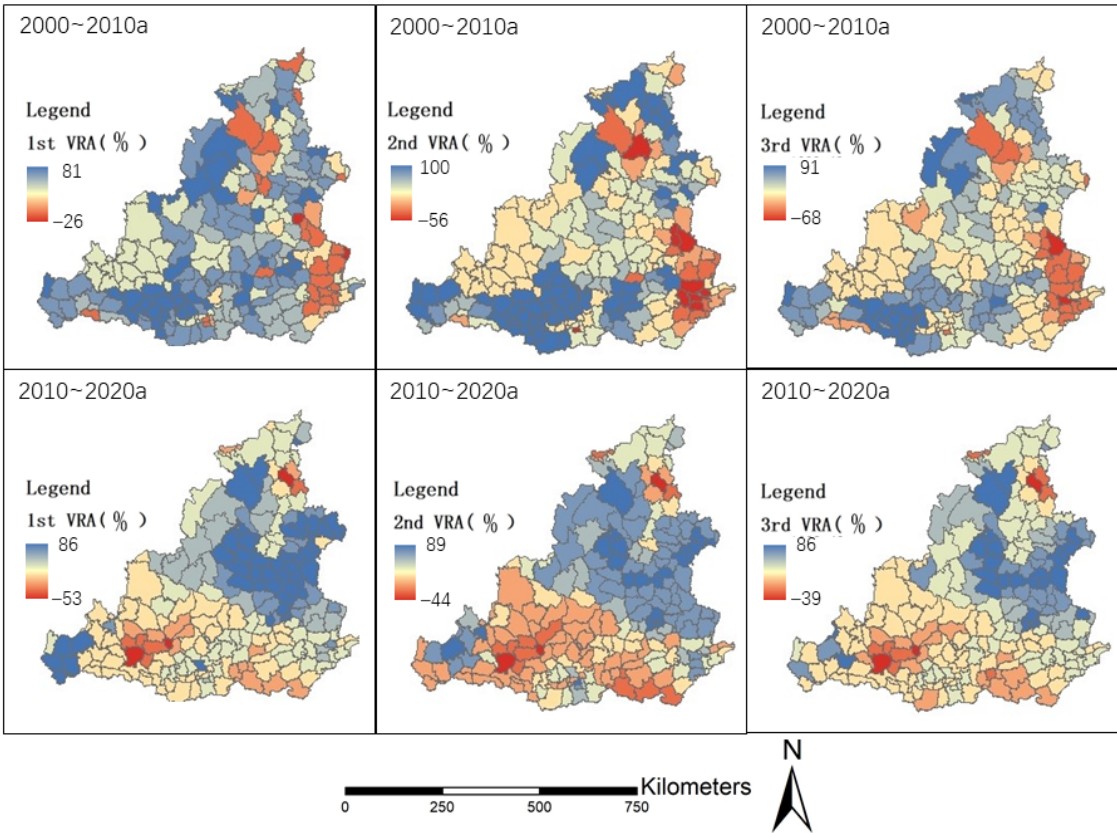

**Figure 6.** Distribution of total soil conservation restoration rate of 226 ERUs.

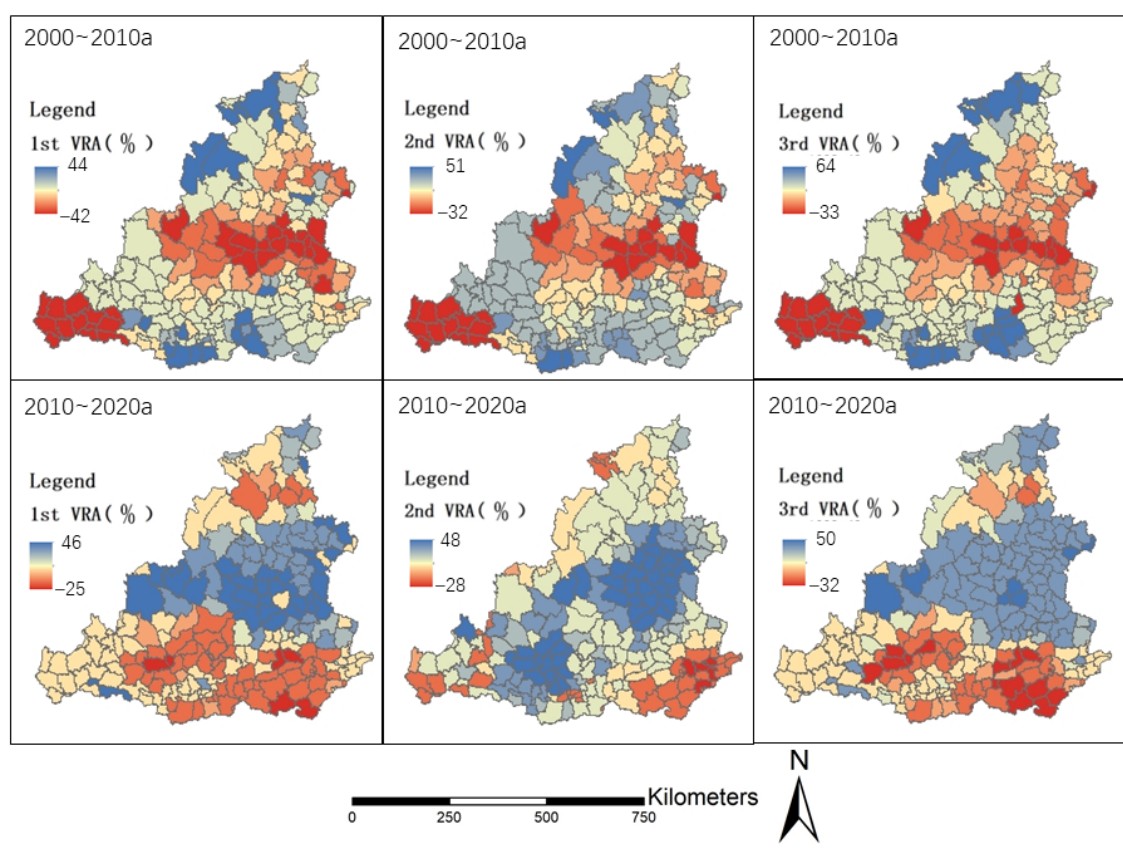

**Figure 7.** Distribution of total water service volume and restoration rate of 226 ERUs.

*3.3. The Balance and Synergy Comparison between the New Vegetation and the Original Vegetation Ecological Service Function*

Vegetation restoration changes the structure and method of land use. It will have a certain impact on the coordination and trade-off of ESs in the VRA. Correlation analysis, cluster analysis, PPF curves, and other methods can verify the trade-off or synergy between ESs. With reference to previous research results, in order to verify the relationship between ESs on the ERU scale, this paper uses correlation to verify the relationship among NPP, SC, and WY. Based on the ERUs, the correlation calculations between the ESs of 2000–2010a and 2010–2020a were carried out in space. In the ArcGIS software, the correlation coefficient between the ESs after restoration was subtracted from the correlation coefficient before restoration, and the time and space of the trade-off and synergy were analyzed in the two periods (2000–2010a and 2010–2020a).

3.3.1. NPP and SC

Before and after recovery in the two periods (2000–2010a and 2010–2020a), the correlation coefficient of the VRA's NPP–SC was positive as a whole (Figure 8), and the change in the correlation coefficient was positive as a whole. The results showed that the NPP–SC relationship of the three VRAs before and after restoration was mainly synergistic as a whole, and the NPP–SC synergistic relationship caused by vegetation restoration was better than that before restoration. During 2000~2010a, the percentages of the ERU with positive changes in the correlation coefficients of the three VRAs were 79%, 81%, and 80%, respectively. Vegetation restoration led to a general enhancement in the NPP–SC synergy (Figure 8). The synergy of the first VRA in ERUs such as Huanglong, Jiangxian, and Dingbian increased significantly; the synergy of the 2nd VRA in ERUs such as Yichuan, Heshui, and Yicheng increased significantly; the ERUs with enhanced synergy of the third VRA were concentrated in the east, and the synergy of ERUs such as Gujiao, Qinyuan, and Qinshui increased significantly. During 2010~2020a, among the three VRAs in the study area, the proportions of ERUs with positive correlation coefficient changes were 77%, 63%, and 73%, respectively. The first VRA increased significantly in ERUs such as Wushen, Yuyang, and Qingjian; the second VRA increased significantly in ERUs such as Tongwei, Zhungeer, and Hongdong; the third VRA increased significantly in Mizhi, and the synergy of ERUs in the counties Zhangxian and Yijinhuoluo increased significantly.

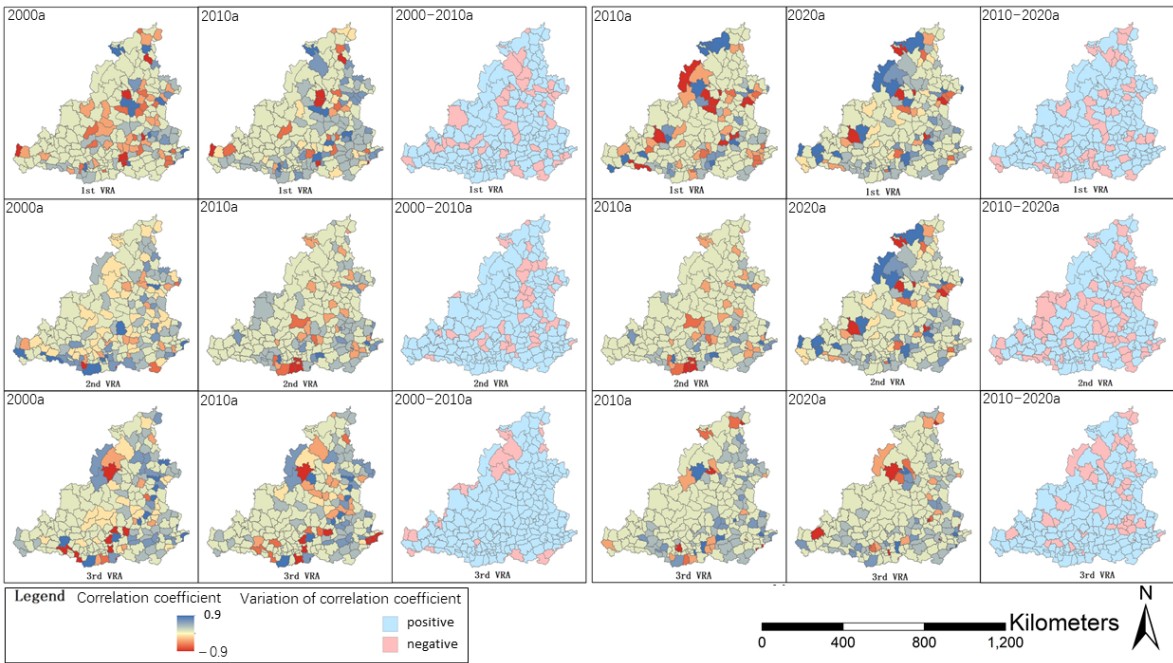

**Figure 8.** The spatial distribution of NPP–SC trade-off and synergy among 226 ecological units.

### 3.3.2. NPP and WY

In the two periods (2000–2010a and 2010–2020a) before and after recovery, the NPP–WY correlations of the three VRAs were negative as a whole. The change in the correlation coefficient is positive as a whole (Figure 9). The results show that the NPP–WY relationship of the three VRAs before and after vegetation restoration is based on the overall trade-off, and vegetation restoration will alleviate the NPP–WY trade-off relationship. During 2000~2010a, among the three VRAs in the study area, the proportion of ERUs showing a weakened trade-off relationship was 94%, 69%, and 81%, respectively. The trade-off relationship of the first VRA in ERUs such as Pucheng, Lingtai, and Dingbian was significantly weakened; the trade-off relationship of the second VRA in Linxian, Xing, Fangshan, and other ERUs weakened significantly, and ERUs with an enhanced trade-off relationship were concentrated in Yuci, Fu County, Kongtong, etc.; the third VRA significantly weakened the trade-off relationship among ERUs in Youyu, Delong, and Salt Lake District. During 2010~2020a, in the three VRAs in the study area, the proportion of ERUs whose trade-off relationship was weakened was 72%, 67%, and 77%, respectively. The trade-off relationship of the first VRA in Jiaxian, Dongsheng, Qianxian, and other ERUs was significantly weakened, and the trade-off relationship of a small number of ERUs in Wuqi, Pingyao, and Gangu increased; the trade-off relationship of the second VRA in Yongji, Huazhou, and Zhenyuan, among other ERUs, weakened significantly; the trade-off relationship among ERUs in Wuqi, Fugu, Mizhi, etc. of the third VRA domain was significantly weakened.

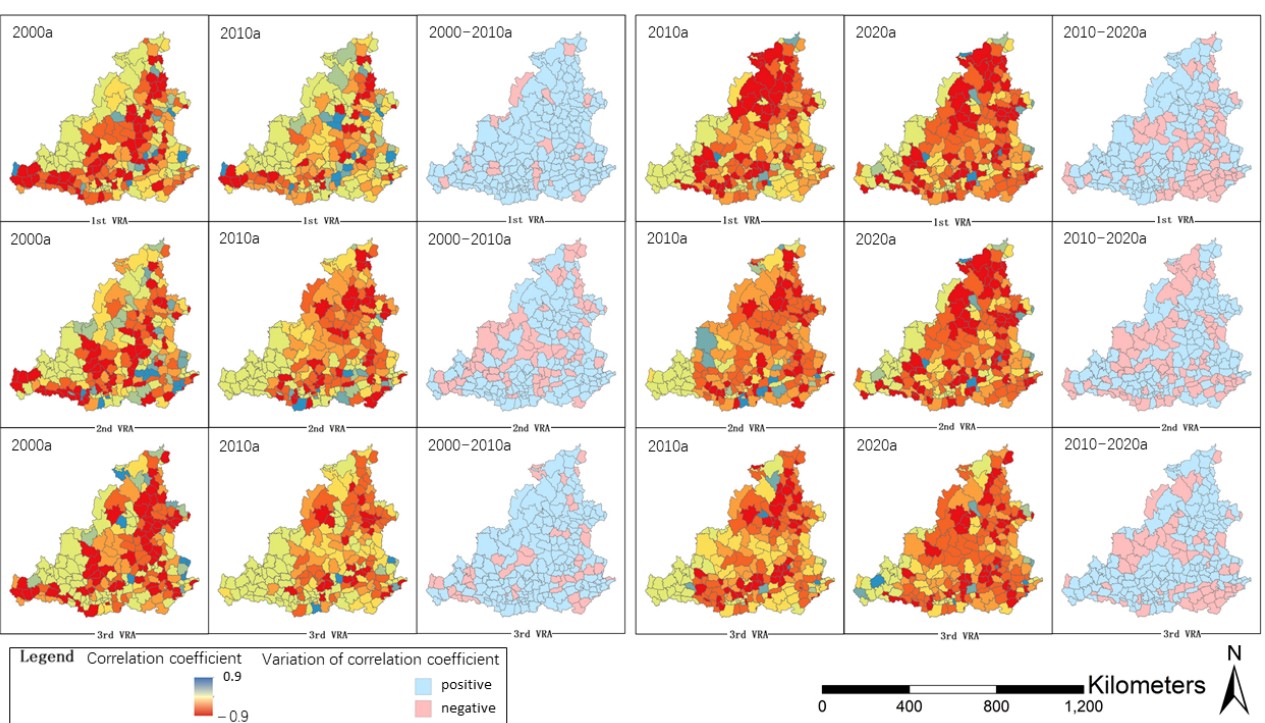

**Figure 9.** The spatial distribution of NPP–WY trade-off and synergy among 226 ecological units.

### 3.3.3. SC and WY

Before and after recovery in the two periods of 2000–2010a and 2010–2020a, the SC–WY correlation coefficients of the three EVRs were negative as a whole, but the correlation coefficient changes were positive. The SC–WY relationship of the three VRAs before and after restoration was based on trade-offs. After vegetation restoration, the overall SC–WY trade-off relationship was alleviated. During 2000–2010a, among the three VRAs (Figure 10), the proportions of ERUs with weakened trade-off relationships were 91%, 71%, and 75%, respectively. The trade-off relationship of the first VRA in ERUs such as Fenyang, Tongwei, and Fangshan was significantly weakened; the trade-off relationship of the second VRA in ERUs such as Huachi, Heshui, and Xiyang weakened significantly; the trade-off

relationship in the third VRA in Youyu and Baishui, among other ERUs such as Wushan, weakened significantly. In 2010–2020a, compared with 2000–2010a, vegetation restoration had a weaker mitigation effect on the SC–WY trade-off relationship, and the proportion of ERUs with a weakened trade-off relationship was 80%, 69%, and 81%, respectively. The first VRA significantly weakened the trade-off relationship among ERUs in Yuyang, Qingjian, and Gangu; the second VRA significantly weakened the trade-off relationship among ERUs in Qishan, Huayin, and Hengshan; the trade-off relationship among ERUs such as Houma weakened significantly.

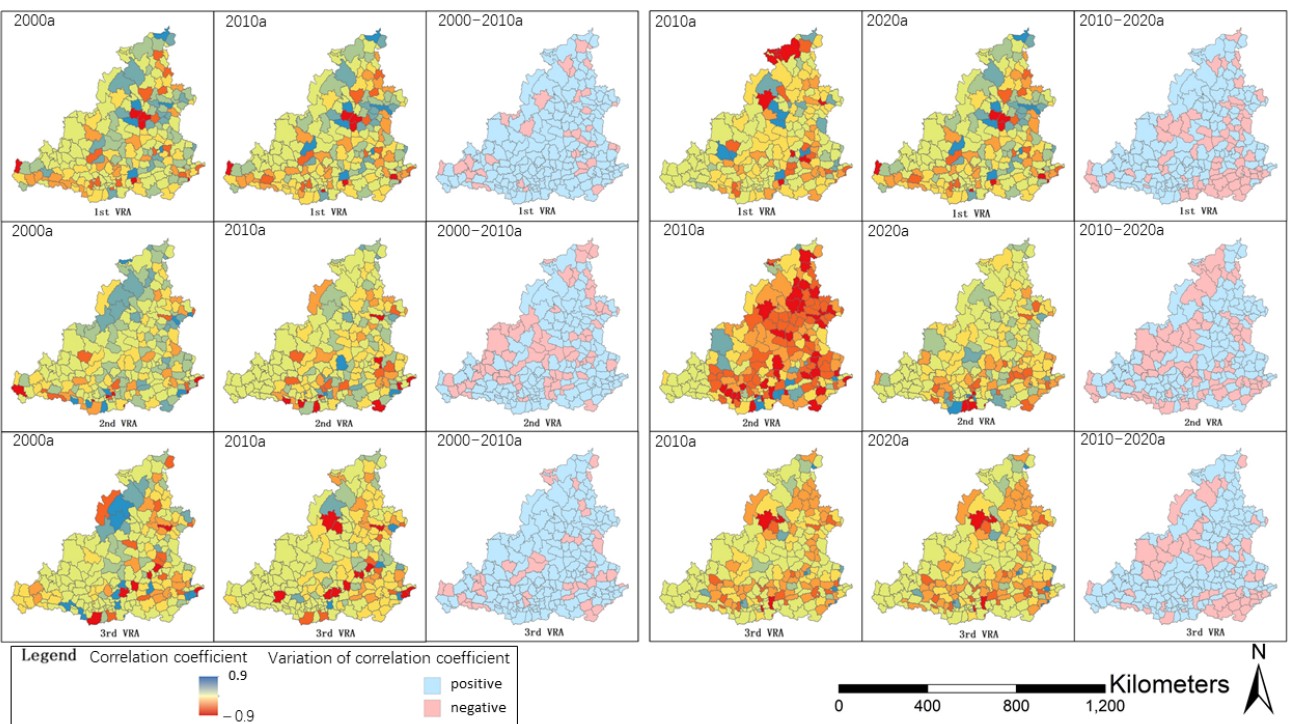

**Figure 10.** The spatial distribution of SC–WY trade-off and synergy among 226 ecological units.

### 3.4. Identification of Factors Affecting Ecosystem Services in Vegetation Restoration Areas

This study selected 12 possible driving factors, including annual precipitation, annual average temperature, NDVI, tree coverage, non-tree vegetation coverage, sand content, soil content, clay content, altitude, surface roughness, GDP, and population density. The factor detector of the geographic detector was used to identify the driving factors of the change in ecosystem services and the trade-off-coordination relationship in the VRA, stratified according to the ERU. The geographic detector was used for multi-level detection, and the Q-value was used to quantify the impact degree. According to the Q value of the driving factors ($p < 0.1$), the influencing factors that pass the *p*-value test and whose Q-value is greater than 0.2 are called the main driving factors; The driving factors that pass the *p*-value test and whose Q-value is less than 0.2 are called important driving forces.

#### 3.4.1. Identification of Factors Affecting Changes in Ecosystem Services in Vegetation Restoration Areas

The changes in the total amount of ecological services in each ERU were hierarchically counted. The change in ecological services was taken as the variable Y, and geographic detectors were used to perform multi-level detection and produce driving force detection results (Figure 11). During 2000–2010a, the average annual precipitation, temperature, and NDVI were the dominant driving factors for the changes in the ecosystem services of the three VRAs. Surface roughness was the main driving factor for the SC changes of the second VRA and the third VRA. During 2010–2020a, the annual average temperature and NDVI were the leading driving factors for the NPP changes of the three VRAs. Surface

roughness was the leading driving factor for the soil conservation changes of the three VRAs. GDP was the leading driving factor for the NPP changes of the second VRA. Soil sand content and soil mud content were the main driving factors for the SC change of the third VRA.

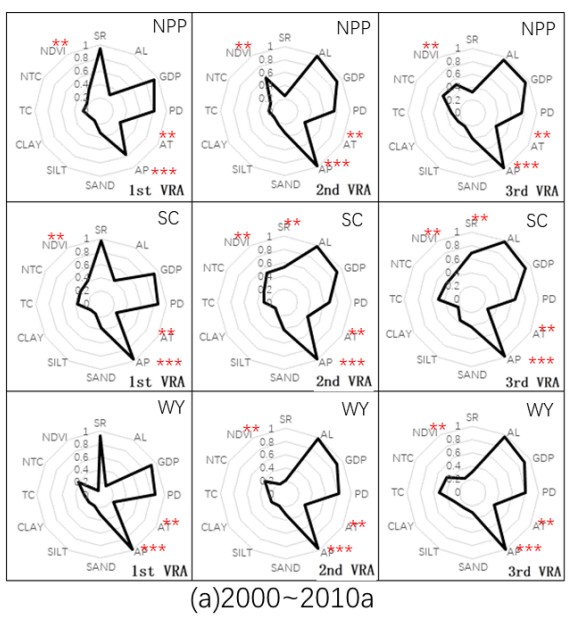

(a)2000~2010a

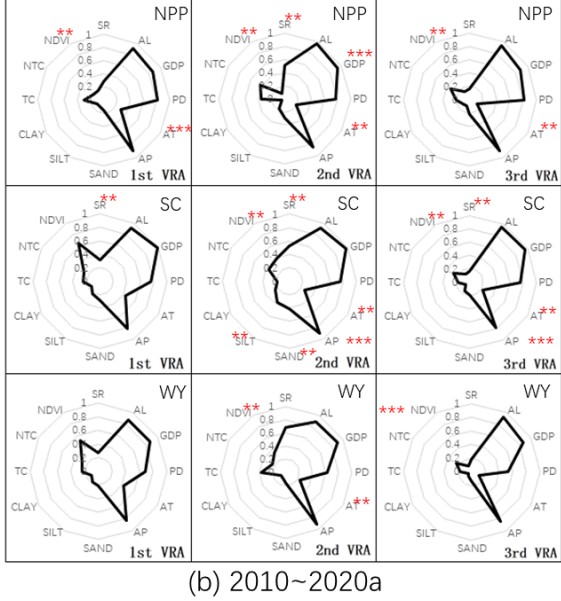

(b) 2010~2020a

**Figure 11.** The larger the q-value of the graph, the more the independent variable will explain the variable to which driving factors belong. The *p*-value represents the degree of significance, and it is very important to require it to be a significant variable at least at the 5% level (*p* < 0.05). They are marked as ***. The other significant variable at least at the 5% level (*p* < 0.01). They are marked as **.

### 3.4.2. Ecosystem Service Trade-Offs and the Identification of Synergistic Influence Factors in Vegetation Restoration Areas

The correlation value of the two ecosystems in each ERUs was hierarchically used as the variable Y, and the geo-detector was used for multi-level detection to detect the driving force changes (Figure 12). In the first VRA from 2000 to 2010a, the air temperature was an important driving factor for changes in the NPP–SC relationship. Surface roughness and NDVI were important driving factors for the changes in the NPP–WY relationship in all three VRAs, and NDVI was an important factor for the SC–WY as well. Clay content was

an important driving factor for NPP–SC of the second VRA and for NPP–WY of the third VRA. From 2010 to 2020a, the annual average temperature and surface roughness were important driving factors for the changes in the three VRAs' ecosystem services. In the third VRA, GDP was the dominant driving factor for NPP–SC and SC–WY.

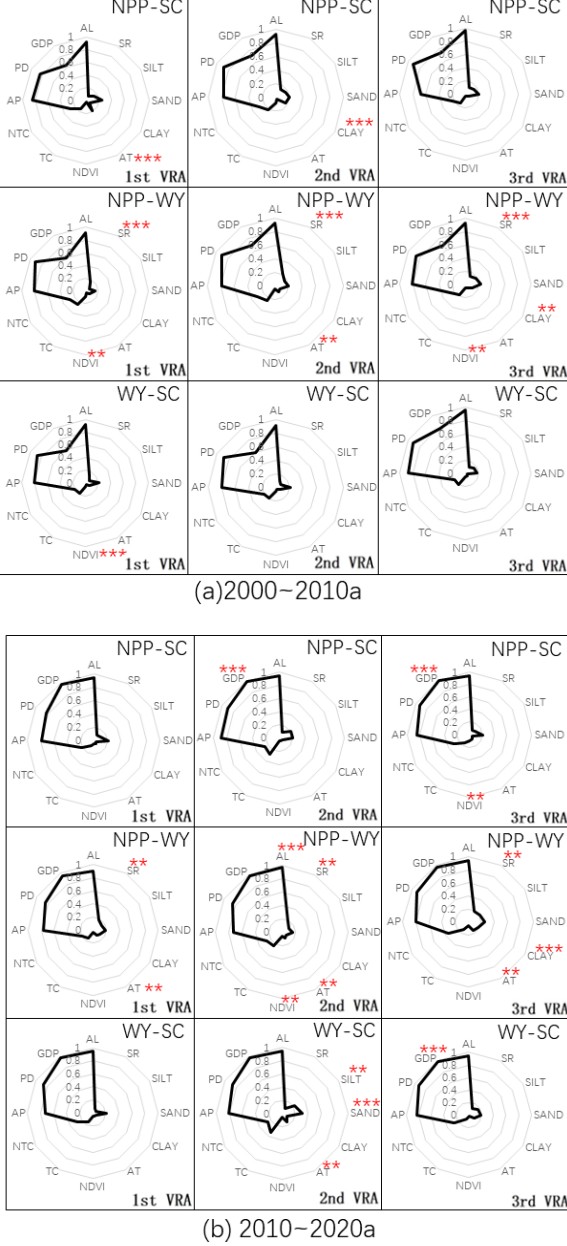

**Figure 12.** The larger the q-value of the graph, the stronger the explanation of the more prominent independent variable with respect to the variable to which driving factors belong. The *p*-value represents the degree of significance, and it is very important to require it to be a significant variable at least at the 5% level ($p < 0.05$). They are marked as ***. The other significant variable at least at the 5% level ($p < 0.01$). They are marked as **.

## 4. Discussion

### 4.1. Differences in Restoration Rates among Ecosystem Services

China is a densely populated country with a small area of arable land. As the population continues to grow, the demand for food continues to increase, resulting in the cultivation of hills and slopes that are not suitable for cultivation to meet the food demand

in areas with scarce arable land. The implementation of the Grain for Green Project changed the land use in the VRA in the middle reaches of the Yellow River and ultimately affected the ecosystem services, consistent with the results of previous studies [51]. The results showed that in the first decade (2000–2010a) and the second decade (2010–2020a) of the implementation of the vegetation restoration plan, the restoration rate of NPP and soil conservation in the restored area as a whole increased, and the restoration rate of water production services decreased.

In 2000–2010a, the total SC recovery rate of the first VRA and the third VRA was higher than that of the second VRA. Different types of vegetation can reduce the damage of soil erosion to varying degrees. A forest canopy can reduce more raindrop kinetic energy compared to grassland. A forest surface cover with a higher density can disperse runoff energy and tree roots. It is more developed than the grass root system and can significantly improve the soil resistance to runoff erosion. Vegetation restoration can prolong the runoff time of slopes, increase infiltration, and reduce the effects of runoff and sediment production. Compared with 2010a, the second VRA had a higher total SC recovery rate in 2020a, and annual precipitation was the dominant factor (Figure 11). According to previous studies [52], the area where the vegetation restoration type is forest had greater evaporation, which led to an increase in local precipitation, and new forests were greatly affected by rainfall erosion, resulting in a lower total SC in the first and third VRAs [53].

NPP removes the carbon fixed by autotrophic respiration during vegetation photosynthesis and is an important indicator for evaluating terrestrial ecosystem productivity and climate change. NPP can be understood as the annual increase in carbon from wood and litter. As more carbon dioxide is fixed by plants in the form of organic matter, NPP increases. In the two periods of 2000–2010a and 2010a–2020a, the total NPP restoration rate of newly added forest was significantly higher than that of newly added grassland (Figure 5). From 2000 to 2010a, in the three VRAs, the NPP increase of the newly added 1 $km^2$ of forest land was $6.78 \times 10^7$ Gc, $2.8 \times 10^7$ Gc, and $4.96 \times 10^7$ Gc; from 2010 to 2020a, under the three restoration models, one square kilometer was added. The increase in NPP of forest land was $1.17 \times 10^8$ Gc, $8.26 \times 10^7$ Gc, and $1.3 \times 10^8$ Gc. The first and third planting restoration methods had better carbon sequestration effects. According to previous studies [54], the impact of forest vegetation on regional climate is more obvious than that of sparse vegetation (such as grassland). As the annual average temperature and annual precipitation in 2000a, 2010a, and 2020a show an increasing trend, more suitable temperature and moisture conditions will promote the growth of vegetation. At the same time, the forest area has a lower albedo, and the vegetation coverage and leaf area index are higher. The maximum transpiration rate will be higher than other types of ecosystems. The newly added woodland and grassland will bring more surface runoff and precipitation into the atmosphere through photosynthesis and transpiration.

In the middle basin of the Yellow River where there is high soil erosion, the reduction of surface runoff can reduce soil loss and play an important role in restoring regional soil conservation ecosystem services [55]. At the same time, the release of steam from plant leaves can improve the temperature and water conditions in the recovery area. In turn, the improvement of temperature and water conditions will promote plant growth and increase the carbon fixation of vegetation [56].

As an important ecosystem service, WY plays an important role in maintaining the stability of an ecosystem. WY can have two effects in space: in the WY area, the hydrological cycle process in the formation of WY affects the carbon cycle, vegetation growth, and other processes; outside the WY area, it affects vegetation. The restoration of regional water production services showed a decreasing trend (Figure 7). New forests and grassland require more water to grow, and surface runoff continues to seep continuously to replenish soil moisture, resulting in a decrease in water volume downstream of the river [57]. The reduction of soil moisture caused by ecological restoration was universal. The amount of soil water storage varies significantly in the middle basin of the Yellow River. Vege-

tation restoration on some slopes can alleviate soil erosion in this area and increase soil water storage.

However, it was not necessary to carry out soil water storage in areas with good soil moisture [58]. Proper afforestation would cause serious damage to deep soil moisture. Decreasing soil moisture will increase the thickness of the dry soil layer. Surface runoff and precipitation will continue to replenish moisture in the dry soil layer, resulting in a decrease in the WY of the VRA. According to the runoff statistics of the Yellow River Sediment Bulletin (2000a, 2010a, 2020a), the runoff difference between Huayuankou and Hekou Town represents the impact of WY in the middle reaches of the Yellow River on the water volume of the Yellow River. In 2000, the water production in the middle reaches of the Yellow River was 17.31 billion cubic meters [59].

This may also confirm the point we made in the introduction that the GFA reduced soil erosion and increased carbon sequestration in the middle Yellow River [60]. However, additional vegetation may reduce water production services and exacerbate drought problems in some regions. In addition, it is clear that the new forests and grasslands in the middle reaches of the Yellow River require more water to grow, resulting in a decrease in water production and are a weak area that requires increased water production services through rational planning of natural ecological projects. This is also consistent with findings on ecosystem services in the middle reaches of the Yellow River [61].

### 4.2. The Impact of Trade-Offs on Vegetation Restoration as an Important Basis for Ecosystem Management

Ecosystem management integrates human values and socioeconomic conditions into ecosystem management to restore and maintain the integrity and sustainability of the ecosystem [62]. The ecosystem is the material basis for human survival, and exploring the structure and function of an ecosystem is required for ecosystem management. ESs can quantitatively evaluate the structure and function of ecosystems, and they are an important hub that links ecosystems and human well-being. The trade-off and coordination of their services are a good measure for formulating ecosystem management objectives and evaluation standards [63]. Vegetation restoration is an important means of ecological restoration. Different vegetation restoration methods lead to differences in the changes in the relationship between ESs. It is important to explore the impact of vegetation restoration methods on the balance and coordination of ecosystems and apply them to the management of ecosystems in the middle stream of the Yellow River [64]. During the two periods of 2000–2010a and 2010–2020a, the interactions between NPP and SC, between NPP and WY, and between SC and WY in the ecological restoration areas in all ERUs were significant ($p < 0.01$).

In the three VRAs during this period, more than 60% of ERUs showed an enhanced synergistic relationship between NPP and SC; at the same time, more than 61% of the ERUs showed an increased balance between NPP and WY, and more than 63% of the ERUs showed a balance between soil conservation and WY. The restriction of WY on soil conservation was weaker than that on NPP. Although suitable climatic conditions and rapid vegetation restoration led to the improvement of the VRAs' NPP and SC, due to climate differences, the hydrological response was significantly different. In arid regions, excessive afforestation will cause vegetation transpiration and soil moisture evaporation, consume soil water resources, and cause WY to decline [65].

NPP–SC correlation coefficient change in more than 60% of the ERUs was Positive. The three VRMs enhanced the NPP–SC synergy, indicating that SC has increased with the increase of NPP, and the two has increased together, the variation in the NPP–WY correlation coefficient of most ERUs is negative [66]. The three VRMs have weakened the trade-off between NPP and WY to varying degrees (Figure 9). Because VRAs were in the recovery stage, their self-regulation ability and stability are poor [67]. There was no complete and stable ecosystem within. The change in the correlation coefficient of NPP–WY for most ERUs was negative. On the whole, the three recovery models had eased the

trade-off relationship between SC and WY (Figure 10). Vegetation restoration strengthens the synergistic relationship between ESs and weakens the trade-off relationship between ESs, which was in line with the goals of ecosystem management [68]. The synergy between a few ERUs in Figure 8 was not strengthened, and the trade-off relationship shown in Figures 9 and 10 was not weakened.

In the next step, the vegetation restoration projects should be adjusted for these ERUs. The most significant synergistic increase occurred in the NPP and SC of the third VRA. The restoration goal of the third VRA was afforestation in barren hills suitable for afforestation [69]. The land-use type before vegetation restoration was grassland, and the soil moisture conditions were good. The first and second VRAs are, respectively, returning farmland to forest and returning farmland to grassland. Before vegetation restoration, it was ditch or sloping farmland, which had poor water and soil retention, resulting in soil moisture and nutrient conditions that were not conducive to vegetation restoration. At the same time, the growth conditions of trees in the third VRA are better than those of the first and second VRA. The strongest trade-off was in the NPP and WY of the first VRA. The growth of trees consumes a high amount of soil water, and the annual evaporation in some restored areas will be greater than the precipitation, and it will continue to absorb water from the soil [70]. Because the Yellow River Basin is located in arid and semi-arid regions, surface runoff is insufficient to supplement the soil moisture [71]. As time goes by, the continuous decrease of soil moisture will become an important factor that inhibits the increase in vegetation NPP. Therefore, using the first VRM and the third VRA in areas with suitable rainfall and soil moisture conditions can effectively increase the synergy between NPP and SC in the region [72]. In areas where precipitation is scarce and soil moisture is low, the 2rd VRM is preferred, so that slow-growing and drought-tolerant herbaceous plants can avoid strengthening the trade-off relationship between NPP–WY and SC–WY production, which leads to the aggravation of drought [73].

There are many studies on the impact of vegetation restoration on ecosystem services in the middle reaches of the Yellow River, most of which are aimed at a single time point or a single ecosystem service [74]. Regarding the impact of vegetation restoration on ecological service functions, the research focuses on evaluating the trade-off-synergy relationship of regional ecosystem services after restoration. At the same time, there is a lack of comparative research on different vegetation restoration methods before and after restoration [75]. Compared with other studies, in this study, a VRA-regulated ecosystem service evaluation framework with ERU as the evaluation unit is proposed. At the regional and ecological restoration unit scales, the changes in ecosystem services and the changes in ecosystem service trade-offs and synergies before and after vegetation restoration were compared [76]. In the context of GFGP, assessing the impact of vegetation restoration on ecosystem services has more practical implications. However, the above evaluation framework is only a reference for decision makers, not a final solution. We will consider more geographic and environmental factors in the future, and by combining the results of this study, we will make more realistic plans for the construction of national ecological projects in the middle reaches of the Yellow River [77].

### 4.3. Prospects and Limitations of This Study

At present, ecosystem research on the middle reaches of the Yellow River focuses on a single time node or a single ecosystem service [78]. The research focuses on evaluating the trade-off-synergy relationship of regional ecosystem services after restoration, and there is a lack of comparative studies on different vegetation restoration methods. It is easy to ignore the scale effect in research [79]. In other words, the current evaluation of the trade-off-synergy relationship based on pixels or regional ecosystem services has no direct guiding significance for the subsequent ecological restoration and ecosystem management in administrative units [80]. Furthermore, in this study, we found that different vegetation restorations resulted in significant differences in the rate of restoration of regulated ecosystem services [81]. This study also has another limitation, the results of

the study are only applicable to the middle reaches of the Yellow River, and the selection of study areas at different scales may have an impact on the results [82]. In the future, we will conduct similar studies at different scales of the research field to explore the applicability of the findings of this study. In addition, if we face a more complex situation in the field of large-scale research, we will choose other ecosystem services (providing services, cultural services) for research, and we will consider introducing ecological process models.

## 5. Conclusions

In this paper, we introduced the VRA's RESs evaluation framework with ERU as the evaluation unit, quantify the impact of three vegetation restoration methods on RESs (including NPP, SC, WY), and used geographic sensors to identify Drivers of regional ecological service restoration. The results showed that, from 2000 to 2020, different vegetation restorations resulted in significant differences in the restoration rates of regulated ecosystem services. Taken together, the three VRMs improved the recovery rates of NPP and SC, while the first and third VRMs decreased the recovery rates of WY. Three types of VRMs will enhance the synergistic relationship between NPP and SC and alleviate the trade-off relationship between NPP–WY and SC–WY to varying degrees. The analysis showed that climatic factors and vegetation coverage affect the regional variation of vegetation restoration, and the coordinated balance of regulating ecosystem services is affected by precipitation, NDVI and GDP. In the current study, the VRA's RESs Evaluation Framework demonstrates great potential for evaluating the benefits and drivers of vegetation restoration, which can be used to evaluate the effects of ecological engineering implementation and serve as a reference for maintenance programs. In this way, scientific advice is provided for the implementation of subsequent vegetation restoration projects and ecosystem management.

**Author Contributions:** D.Y. and G.W. contributed to the conception of the study; T.N. and G.W. performed the experiment; Q.Y., T.N. and G.W. contributed significantly to analysis and manuscript preparation; Q.Y., T.N. and G.W. performed the data analyses and wrote the manuscript; Q.Y., T.N. and G.W. helped perform the analysis with constructive discussions. All authors have read and agreed to the published version of the manuscript.

**Funding:** This research is supported by National Natural Science Foundation of China (No. 42071237) and the Youth Science Foundation of National Natural Science Foundation of China (No. 42001211).

**Institutional Review Board Statement:** Not applicable.

**Informed Consent Statement:** Not applicable.

**Data Availability Statement:** The data presented in this study are available on request from the corresponding author.

**Conflicts of Interest:** The authors declare no conflict of interest.

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
