# Peer review of "Regulated Ecosystem Services Trade-Offs: Synergy Research and Driver Identification in the Vegetation Restoration Area of the Middle Stream of the Yellow River"

_remotesensing, doi:10.3390/rs14030718_

Round 1

Reviewer 1 Report

The study analyzed ecosystem services in the vegetation restoration area in the middle stream of the Yellow River for 20 years (2000–2010a and 2010–2020a) considering Net Primary Productivity, soil conservation, and water yield.

I think the methodology is developed good indicators to analyze the ecosystem services variations and it could be applied in other studies therefore it is good work

The introduction, materials and methods are well developed and give full information about the topic

I have just one doubt about the ecosystem service restoration rate

You reported; ??recove = (??After – ??Before)/ ??After × 100%. Could be more convenient and correct to calculate it like (??After – ??Before)/ ??Before × 100%? (A)

The results, maybe you can color the values in the diagonal to read better the table. I think that the diagonal are represents the quantity of each class that does not change.

In the results, maybe you can color the values in the diagonal of tables 2 and 3 to read better the table. I think that the diagonals represent the quantity of each class that does not change in the time

You could review the results in consideration of point A if  you will change the formula

Discussion and conclusion are ok. they are coherent with the aims and results

Author Response

Dear reviewers:

Thank you for your comments on our manuscript entitled “Regulated ecosystem services trade-offs, synergy research and driver identification in the vegetation restoration area of the middle stream of the Yellow River” (ID: remotesensing-1540226). Those comments are very helpful for revising and improving our paper, as well as the important guiding significance to other research. We have studied the comments carefully and made corrections which we hope meet with approval. The main corrections are in the manuscript and the responds to the reviewers’ comments are as follows.

Point 1: ??recove = (??After – ??Before)/ ??After × 100%. Could be more convenient and correct to calculate it like (??After – ??Before)/ ??Before × 100%?

Response 1: We have modified Equation (1) and recalculated the recovery rate in the Results section.

Point 2: The results, maybe you can color the values in the diagonal to read better the table.

Response 2: We have revised Tables 2 and 3, and marked the unchanged values in red.

Reviewer 2 Report

Dear authors, after reviewing your manuscript I would like to make a few simple comments which I suggest to be implemented in order to proceed with the acceptance of your article:

1.- Please add in Line 10 the abbreviature ES after Ecosystem Services.

2.- In the introduction you talk about support services. Nowadays, according to the latest publications, these services are considered as part of the regulatory services. Please use this classification and include support services as a type of regulatory services.

3.- The introduction is very well developed, but I think that in some parts they specify too much detail with respect to previous studies. Please summarise the information concisely.

4.- Study area: please specify the country of the study area.

5.- Please improve Figure 1. Enlarge this Figure.

6.- Data source: please remove the links, these links should be references.

7.- Please specify all equations with their units.

8.- Please enlarge figures in the results of the manuscript.

9.- You use a p-value < 0.1, why?, this must be specified because in Natural Sciences, normally, a p-value < 0.05 is used.

10.- Please add more references in the discussión.

11.- I´m not sure about this comment, but I think that Remote Sensing ask for the references in a number format and not specifying the surname of the authors. Please verify.

Kind regards,

Author Response

Dear reviewers:

Thank you for your comments on our manuscript entitled “Regulated ecosystem services trade-offs, synergy research and driver identification in the vegetation restoration area of the middle stream of the Yellow River” (ID: remotesensing-1540226). Those comments are very helpful for revising and improving our paper, as well as the important guiding significance to other research. We have studied the comments carefully and made corrections which we hope meet with approval. The main corrections are in the manuscript and the responds to the reviewers’ comments are as follows.

Point 1: Please add in Line 10 the abbreviature ES after Ecosystem Services.

Response 1: We have added abbreviations after ecosystem services on line ten

Point 2: In the introduction you talk about support services. Nowadays, according to the latest publications, these services are considered as part of the regulatory services. Please use this classification and include support services as a type of regulatory services.

Response 2: In the introduction, we made changes to support services to classify them as regulated services.

Point 3: The introduction is very well developed, but I think that in some parts they specify too much detail with respect to previous studies. Please summarise the information concisely.

Response 3: We have simplified the details of previous research in the Introduction.

Point 4: Study area: please specify the country of the study area.

Response 4: In Figure 1, we have marked the countries where the training area is located.

Point 5: Please improve Figure 1. Enlarge this Figure.

Response 5: We have modified Figure 1 and enlarged Figure 1.

Point 6: Data source: please remove the links, these links should be references.

Response 6: We have removed the connection in the quantity source.

Point 7: Please specify all equations with their units.

Response 7: We have added units to the formula.

Point 8: Please enlarge figures in the results of the manuscript

Response 8: We have zoomed in on the values in the Results section.

Point 9: You use a p-value < 0.1, why?, this must be specified because in Natural Sciences, normally, a p-value < 0.05 is used

Response 9: We have re-run the p-value test (p<0.05).

Point 10: Please add more references in the discussión

Response 10: We have added more references to the Discussion.

Point 11: I´m not sure about this comment, but I think that Remote Sensing ask for the references in a number format and not specifying the surname of the authors. Please verify.

Response 11: We have modified the format of the bibliography as required by Remote Sensing

Reviewer 3 Report

The article is interesting and could make an important contribution to the field, but unfortunately in its current form the manuscript lacks research depth and requires more development with respect to the discussions and conclusions, which are too descriptive, and lack the analytical touch required to emphasize the contribution of the study to the theoretical advancement of the field, and the international exposure required for publication in an international journal. In addition, a conceptual error requires reconsidering the title, which is too general, and perhaps other parts of the manuscript as well. Detailed comments are provided in each case.

The main problem of the article is a conceptual one. As the authors correctly cite the literature, ecosystem services can be classified as supply services, regulation services, cultural services, and support services (line 36). However, although the article is titled "ecosystem service trade-offs...", leaving the impression that all types of ecosystem services are dealt with, the study only estimates the supply services. Therefore, they should rewrite all essential parts of the paper, including its title, to clearly state that they are not dealing with all types of ecosystem services.

Figure 1 shows the inability of authors to write up research. This is an article for an international journal, and not a report for the national authorities. The map should not be presented as if China is the only country of the world, but show also the neighboring countries with their names, so that a Brazilian researcher could understand it too.

The discussions should include (A) the significance of results - what do they say, in scientific terms; (B) the inner validation of results, against the study goals or hypotheses; (C) the external validation of results, against those of similar studies from other countries, identified in the literature; (D) the importance of the results, meaning their contribution (conceptual or methodological) to the theoretical advancement of the field; (E) a summary of the study limitations and directions for overcoming them in the future research. Out of these, only the significance of results is presented; the missing elements should also be developed.

Conclusions are not sufficiently broad in scope, and lack research depth, pertaining only to the case study and being in fact just a summary of the main findings. Conclusions are meant to deliver a scientific message, far away beyond the case study, to the entire scientific community, making a clear contribution to the theoretical (conceptual or methodological) development of the field.

Author Response

Dear reviewers:

Thank you for your comments on our manuscript entitled “Regulated ecosystem services trade-offs, synergy research and driver identification in the vegetation restoration area of the middle stream of the Yellow River” (ID: remotesensing-1540226). Those comments are very helpful for revising and improving our paper, as well as the important guiding significance to other research. We have studied the comments carefully and made corrections which we hope meet with approval. The main corrections are in the manuscript and the

responds to the reviewers’ comments are as follows.

Point 1: The main problem of the article is a conceptual one. As the authors correctly cite the literature, ecosystem services can be classified as supply services, regulation services, cultural services, and support services (line 36). However, although the article is titled "ecosystem service trade-offs...", leaving the impression that all types of ecosystem services are dealt with, the study only estimates the supply services. Therefore, they should rewrite all essential parts of the paper, including its title, to clearly state that they are not dealing with all types of ecosystem services.

Response 1: This paper is aimed at the impact of vegetation restoration on regulating services in the middle reaches of the Yellow River. We have revised important parts of the paper, including the title, abstract and introduction, etc.

Point 2: Figure 1 shows the inability of authors to write up research. This is an article for an international journal, and not a report for the national authorities. The map should not be presented as if China is the only country of the world, but show also the neighboring countries with their names, so that a Brazilian researcher could understand it too.

Response 2: We have modified Figure 1 to label the countries where the study area is located and the surrounding countries.

Point 3: Conclusions are not sufficiently broad in scope, and lack research depth, pertaining only to the case study and being in fact just a summary of the main findings. Conclusions are meant to deliver a scientific message, far away beyond the case study, to the entire scientific community, making a clear contribution to the theoretical (conceptual or methodological) development of the field.

Response 3: We have substantially revised the Discussion section to externally validate the Results section with results from similar studies in other countries identified in the literature. We illustrate their contribution to the theoretical advancement of the field. We summarize study limitations and directions for overcoming these limitations in future research.

Round 2

Reviewer 2 Report

Dear authors, thank you very much for your answers. I recommend the acceptance of the manuscript.

Reviewer 3 Report

The authors have properly and fully addressed all comments, and as a result the manuscript increased its research depth and addresses a broader international community. I do not have any further comments, and recommend the publication of manuscript in its revised form.

This manuscript is a resubmission of an earlier submission. The following is a list of the peer review reports and author responses from that submission.